# The Importance of Measuring Local Governments' Information Disclosure: Comparing Transparency Indices in Spain

**Juan-Carlos Garrido-Rodríguez \*, Marta Garrido-Montañés, Germán López-Pérez and Elisabeth Zafra-Gómez**

Department of Accounting and Finance, University of Granada, 18071 Granada, Spain
\* Correspondence: jcgarrido@ugr.es

**Abstract:** Transparency is considered a key element for developing a reliable government; it is the ability that all public entities have to provide access to all their information. The main objective of this work is to test the differences in the transparency of municipalities between the main indices and a new index made following the regulatory advances in this area at the national level. Called BTI, this new index aims to measure in the best possible way the degree of compliance of Spanish municipalities, through two dimensions (depth and breadth), which measure the quality and the quantity of the information. This work makes a comparison of the BTI with three of the main existing transparency indexes in Spain, to a sample of Spanish provincial capitals. The results obtained reveal clear differences between the indices, where three of the four indices analyzed show that most municipalities only fulfil the Transparency Act at the minimum level, with only a few municipalities reaching the maximum rating range. This work concludes that the BTI index is more demanding than the rest of the indexes. However, ITA stands out with a higher average score than the rest, which indicates that transparency portals are designed to obtain a good score in this index, being less objective. Finally, this paper remembers the importance of having an objective tool to measure transparency, as it can show notable differences with respect to reality.

**Keywords:** transparency; municipalities; regulation; index; portal

## 1. Introduction

Nowadays, transparency has become a fundamental element in developing a reliable and responsible government. As a consequence of this, there has been an increase in citizens' demand for information from public institutions, forcing them to adapt accordingly. Thus, the concept of transparency could be defined as a first step to achieve the objectives of accountability based on the information that public administrations make available to society [1].

According to [2], "a good government must be seen to be real", so we can consider that transparency develops links between institutional trust, political trust, and social trust. There is no universal transparency regulation, but instead each country has developed its own regulation, which implies a great diversity of regulations related to transparency and access to information [3]. In this way, there are several ways to measure transparency and organizations that make indices to evaluate a public bodies' degree of transparency. In Spain, the best known is the governmental organization called Transparency International Spain, which has prepared such an index since 2008 on an annual basis in the 110 Spanish municipalities with the largest population. Other indices to be highlighted are the Infoparticipa Map, which is one of the most relevant in the field of transparency, and its methodology is based on an evaluation of the transparency compliance index of corporate websites by applying a series of indicators adapted to the law and the Decalogue of Good Practices. Additionally, there is the DYNTRA index, which works on the measurement and management of open government in organizations.

In this context, the aforementioned indices have different limitations, related above all to the quality of the information disclosed. Thus, the present paper presents the Bidimensional Transparency Index (BTI), which allows us to break down the information into two dimensions, breadth, and depth, thus allowing us to analyze the quantity and quality of the information disclosed by local entities.

The second objective of this work is to compare the data obtained from the BTI with three other transparency indices—the ITA, DYNTRA, and the Infoparticipa Map—comparing the results obtained for the Spanish provincial capitals and observing the differences in scores with respect to the BTI.

Our scientific contribution is to provide a new index, which brings a new form of measuring transparency, in terms of quality and the quantity of information, so that all citizens can find out any issue related to the management and decisions that the city council is carrying out. We also expect to present an index that treats the information in an adequate form and is a more exact graduation of compliance than the existing ones.

Thus, this paper is structured as follows: the next section develops a theoretical framework related to the concept of transparency; the development of the regulations of transparency law at the national and international levels; and the different ways of measuring the degree of transparency. This is followed by the methodological proposal and the results obtained by the comparison of the different indices and, finally, the conclusions reached with the development of this work.

## 2. Theoretical Approach

### 2.1. A Review of the Transparency Concept

In recent years, as a result of various socio-political crises that have occurred in society due to multiple cases of corruption, and recent major economic recessions, transparency has become a fundamental element of accountability and anti-corruption. Public entities have been adapting to the citizens' demand for public information [4–7] with the aim of increasing public confidence and the legitimacy of the decisions made by public entities [8]. Thus, there is a broad discussion in the literature on the different elements of this concept that should be considered [8–10].

The term "transparency" is defined as the quality of a government, company, organization, or person, to be open with regard to the disclosure of information, actions, or plans [11]. On the other hand, some authors define this concept as a public value that requires public agents to provide their citizens with information on how and why they make decisions, in order to mitigate the risks of corruption [12,13]. Ref. [14] defines it as a virtue in public management that pursues a reliable and responsible government, while [15] emphasize that transparency connotes openness and the absence of opacity, which according to [16], has a great influence on democratic governance to prevent the abuse of power.

However, one of the most relevant definitions that gathers many current elements is the one developed by [17], who considered transparency to be: "*the publicity of all the acts of government and its representatives to provide civil society with relevant information in a timely, useful and comparable way and in an accessible format*". Thus, transparency can be considered a virtue in public management and policymaking, as well as an important democratic value that seeks to pursue the trust and accountability of a government [7,18]. Thus, transparent conduct by government leads to the reduction of information gaps between government and citizens [5,18], whereas the absence of transparency reduces trust on the part of citizens, making them feel excluded from public management and governance [17].

On the other hand, there is some controversy in the definition of the concept of transparency, where it can be characterized as "nebulous" [19], not having an accepted definition due to the diversity of uses it has had among practitioners, politicians, and academics in different fields [20].

According to [21], transparency is a first step towards the accountability of those responsible for public entities, with the latter having to inform citizens about their projects

and the fulfilment of objectives, facilitating forms of participation and development for their achievement. In this sense, some authors identify accountability, transparency, and citizen participation, as the three key elements to recovering the legitimacy of public entities, in addition to identifying them as key tools to improve corruption control [12,22,23]. In this way, we can speak of transparency as synonymous with understanding public policies and increasing trust in the public sector, which leads to a reduction in corruption [13,24,25].

Thus, the essential axes in the public sphere must be based on transparency, access to public information, and good governance standards, so that the action of public entities is subject to scrutiny and citizens can know and have information on how they dispose of public funds and how decisions that directly affect them are made; this is the beginning of a process in which information and participation in public authorities are demanded [26].

Globally, this fact has increased in importance in transparency and information disclosure. In this way, transparency refers to the possibility that the subjects affected by it have to consult the real information of public bodies [13,27]; it is a key concept that determines the responsible consideration of a government, and it is an essential element in determining good governance, which leads to more efficient development of resources [28,29] and higher economic growth [30,31].

In addition, thanks to the information that citizens obtain from public entities, society can participate in the actions of public entities, so it can be considered that transparency determines a dependence between the administration and the population: the "active participation" of citizens in which they influence the actions to be carried out, defining evaluation criteria and decision-making objectives, which optimizes efficiency [13,24,32].

This increased importance of transparency and the disclosure of public information has led to the approval of rules whose main objective is to regulate this principle [33], with city councils being the institutions that should attach the most importance to it, since they are the first level of citizen participation [34].

### 2.2. The Development of the Transparency Act in Spain

As mentioned above, transparency has become an essential element when formulating and managing public policies in democratic governments, which boosts citizens' trust and participation [14]. To contribute to this objective, Spain published the Law 19/2013 of 9 December on transparency and access to public information and good governance; however, this was not the first time that reference was made in the regulatory field to citizens' access to information.

The rights and duties of transparency and access to information burst onto the world scene in Sweden, a country that in 1766 published the first direct state transparency regulation and inspired countries such as the United States, Norway, France, the United Kingdom, and Germany, to subsequently publish their own regulations [35]. The implementation of these laws has reduced the information asymmetry between governments and citizens [18], which has led to a wave of freedom of information that has spread to 115 countries around the world that, by 2017, had already passed their own freedom of information laws [14].

According to [35], the first references to transparency can be found in the "Declaration of the Rights of Man and the Citizen" of 1789; he considers it the germ of global declarations and places special emphasis on Articles 14 and 15, where allusion is made to citizens' right to the verification, contribution, and acceptance of public activity, as well as the right to request information about the management of public agents. On the other hand, ref. [36] alludes to Article 19 of the Universal Declaration of Human Rights of 1948 as a pioneering reference for establishing that: "*everyone has the right … to seek, receive and impart information*" [37].

At the European level, the Declaration of the Committee of Ministers of the Council of Europe on the Freedom of Expression and Information, adopted in 1982, can be

highlighted, as well as Articles 6, 8, and 10, of the European Convention for the Protection of Human Rights and Fundamental Freedoms [38]. In addition, there are different directives on transparency in specific topics, such as the Council Directive of 7 June 1990 on the freedom of access to environmental information [39,40], as well as the Council Directive 2011/85/EU of 8 November 2011 on the requirements applicable to the budgetary frameworks of the Member States, which devotes an entire chapter to the transparency of public administrations' finances. On the other hand, Directives 2014/23/EU, 2014/24/EU, 2014/25/EU, and 2014/55/EU, refer to transparency in public procurement [41]. More recently, after the approval of transparency laws in several member countries, Directive 2019/1024 of the European Parliament and of the Council of 20 June 2019 was published, approved to promote the openness of public information and legal certainty [42].

In Spain, the general principle of transparency should preside over the operation of public administrations in their relations with citizens [43,44]. Therefore, Law 19/2013 of 9 December 2013 on transparency and access to public information and good governance was published at the end of 2013, which came to regulate the historically unattended matter of transparency and access to information in the Spanish public sphere [45]. The principle of transparency, provided for in Article 105(b) of the Spanish Constitution, had been developed until then in general terms in Article 37 of Law 30/1992 of 26 November 1992 on the Legal Regime of Public Administrations and Common Administrative Procedure, and the Law 27/2006 of 18 July 2006, which regulates, among others, access to environmental information. In addition, the Law 37/2007 of 16 November on the reuse of public sector information also contemplated this principle, along with sectorial regulations that include publicity obligations in each field [33].

According to [46], Law 19/2013 has a triple scope:

- ⅃ First, it increases transparency in public activity through active publicity obligations for all administrations and public entities.
- ⅃ Secondly, it recognizes access to information as a right of broad subjective and objective scope.
- ⅃ Finally, it establishes the obligations of good governance to be fulfilled by public officials and the legal consequences deriving from their non-compliance, thus becoming a requirement of responsibility for all those who carry out public activities with public relevance.

Before the approval of this act, there were already several transparency evaluation bodies in Spain; however, the real boom of this principle arose after the approval of this regulation and the creation of the Council for Transparency and Good Governance, with its corresponding equivalent bodies in the autonomous communities [47]. At present, 16 autonomous communities have already developed a transparency law, while one of them has a draft law. In the case of the Autonomous Cities of Ceuta and Melilla, only the latter has a specific regulation on transparency and access to public information.

At the municipal level, the power to develop their own regulations on transparency lies with the municipalities through local transparency ordinances. Although few municipalities have developed their own ordinance regulating this principle, we can take as an example those of Granada, Seville, Cordoba [48], Zaragoza, Madrid, and Valencia.

### 2.3. How Can Transparency Be Measured?

As a result of the publication of regulations governing the principle of transparency, the need has arisen to create tools to measure the degree of transparency of a public entity, and the most widely used form has been the creation of indices that manage to measure the transparency of municipalities based on a given set of indicators. These are usually based on current legislation, which in many cases are structured in rankings from higher to lower levels of transparency [49]. This has led to an increase in competitiveness among municipalities as they aim to move up the ranking by improving their image [31].

At the Spanish level, the following transparency indices are worth mentioning:

1. **The Transparency Index of Municipalities (ITA):** this is an index prepared by the non-governmental organization called Transparency International Spain, which measures the transparency of the websites of the 110 municipalities with the largest number of inhabitants, through 80 dichotomous indicators separated into six different areas. This index began to be elaborated in 2008 and had done so uninterruptedly [8] until 2017, when the last ranking was published. Among the objectives it aims to fulfil, the following stand out: the promotion of the informative culture of the city councils themselves, achieving an increase in the level of useful and important information they offer to citizens and society; fostering a greater approach of the city councils to citizens; and promoting an increase in the information they receive from local corporations (situations, activities, benefits, and services to which citizens can have access). On the other hand, the six areas measured are classified by the type of information and are as follows:

   (a) Information on the municipal corporation.
   (b) Relations with citizens and society.
   (c) Economic and financial transparency.
   (d) Transparency in the contracting of services.
   (e) Transparency in matters of urban planning and public works.
   (f) Right of access to information.

2. **Dynamic Transparency Index (DYNTRA):** the DAM index (DYNTRA of municipalities), is responsible for evaluating and analyzing municipalities with at least 15,000 inhabitants, using 142 dichotomous indicators in six areas [50]:

   (a) Municipal transparency.
   (b) Citizen participation and collaboration.
   (c) Economic-financial transparency.
   (d) Procurement of services.
   (e) Urban planning and public works.
   (f) Open data.

   This index allows real-time measurement, generating a municipal transparency ranking that is in continuous development, since, by including new indicators, the classification changes automatically. It is a collaborative platform whose objective is to measure the public information of public administrations, governments, political parties, elected officials, and different social actors, in an efficient, transparent, and open manner. It is managed by DYNTRA, an international non-profit organization based in Brussels, Belgium [47]. In this sense, this index allows the stakeholders to co-evaluate the level of transparency of a municipality.

3. **Infoparticipa Map:** it was created in 2012 to evaluate the information offered by municipalities on their websites and to obtain more transparent and higher-quality communication, promoting improvements that favored both each of the organizations and the public sector as a whole. What began by analyzing the websites of 947 municipalities in Catalonia ended years later with the help of other researchers, with the evaluation of communities such as Madrid, Andalusia, Aragon, and the autonomous cities of Ceuta, and Melilla, thanks to three R + D + i projects of the Ministry of Economy and Competitiveness. It has also been adapted for application in some Latin American countries. Initially, 41 questions were used, grouped into four blocks: who are the policymakers; how do they manage collective resources; how do they report on management; and what elements do they offer for participation? The Infoparticipa Map carried out the evaluations with 52 indicators in 2015, moving to 48 in 2019, and in 2020 (the latest data obtained), again with 52 indicators to evaluate the city councils [51].

4. **Transparency Evaluation and Monitoring Methodology (MESTA):** developed in 2016, it makes it possible to score all types of administrations according to their own

characteristics and transparency standards. It is the first official transparency evaluation methodology of the Spanish state and is carried out by the Council for Transparency and Good Governance and the Evaluation and Quality Agency in three phases: exercise of the right; processing of the request for access to public information; and completion as a response to the request for public information. Each phase has different criteria (from 14 to 21), and a compliance scale is used for its analysis. An important aspect to note about this index is that it measures two levels of openness in public data: on the one hand, it evaluates the degree of mandatory compliance with the Transparency Law, and on the other hand, it evaluates the quality of transparency with voluntary indicators, which are added to the mandatory ones and are extracted from more complete regulations [52].

Thus, in this context, in the report prepared by Transparency International Spain in 2017, the ITA evaluated the 110 municipalities with the largest populations in Spain, and it can be seen that the average score reaches 90 points (out of 100), compared to 85.2 reached in 2014, and that 25 municipalities reach the maximum score. However, the optimism generated by Transparency International Spain's transparency index should be questioned if we analyze in more detail the quality of the data [12] in that these are not transparent, static, and closed.

Many of these indices are criticized for not making homogeneous measurements among themselves [52], in addition to the fact that there is no graduation of the indicators because they can only be expressed if the information of each one appears; in other words, the indicators are made with dichotomous answers. Furthermore, according to [1], the contents to be evaluated within each index are known in advance by the municipalities evaluated, which causes public managers to focus on such contents and adapt the information published in this index, thereby achieving high results with respect to it and the impossibility of detecting poor performance. This is a fact that can be contrasted by checking how the results obtained from the ITA from 2014 to 2017 go from 19 to 25 municipalities with the highest score. Finally, Ref. [51] exposed that the result obtained from each municipality after the law was approved does not concern whether it is applied or not, since there is no sanction if it is not complied with, in addition to the lack of resources to evaluate compliance and the law's disappearance from the media agenda.

## 3. Data and Methodology

### 3.1. Proposed Methodology to Measure the Degree of Transparency of a Municipality: The Bidimensional Transparency Index

Due to the limitations that the previously described indices possess (see above), in this paper we intend to present a new transparency index developed by [49], which we will call the Bidimensional Transparency Index (BTI). It is based on the requirements established within the Law 19/2013, and is developed in two dimensions: breadth and depth.

Firstly, the breadth is the number of indicators by which this index is formed and how they are distributed. Our index is made up of a total of 20 indicators (see Supplementary Materials File S1), encompassing six areas of evaluation, which are as follows:
(a) Institutional and organizational information: 5 indicators.
(b) Information on senior management and those exercising maximum responsibility: 2 indicators.
(c) Information on planning and evaluation: 2 indicators.
(d) Information of legal relevance: 5 indicators.
(e) Information on contracts, agreements, and subsidies: 4 indicators.
(f) Economic, financial, and budgetary information: 2 indicators.

On the other hand, the depth focuses on the content of each of the indicators. In order to prepare the index, we wanted to give greater depth to each indicator, moving away from the idea of having dichotomous indicators. Thus, each of the indicators has a

structure for its evaluation, where first a series of requirements to be met is established, which depend on each indicator, with the score varying between 0 and 3 as these requirements are met. These requirements have been elaborated based on the content of Law 19/2013 and [33].

When a municipality does not meet any of the requirements established in the indicator, it is scored as 0. As the municipality meets some of the requirements established by the indicator, the values of 1 and 2 will be assigned, leaving the value of 3 for total compliance with the requirements. In this way, the level of transparency is graded, providing depth to the indicator in particular, and to the index itself, when the result of all the indicators is obtained. In addition, each value has an identifying color associated with it, following a "traffic lights" evaluation system [53], with purple for 0, red for 1, yellow for 2, and green for 3.

This allows us to obtain the level of transparency in each of the areas to be evaluated; given that the law itself does not establish anything regarding the importance of each of the obligations imposed, it has been decided to give the same importance to each of the indicators within each area, and the score for each area can be obtained as follows:

$$BTIarea_i = \frac{\sum score\,obtained\,in\,the\,indicators\,of\,the\,area}{maximun\,possible\,score\,of\,the\,area} \times 100$$

where *area$_i$* represents the area of information which is evaluated (from *a* to *f*, shown above).

Just as with the indicators, the score obtained in each area is associated with an identifying color with a "traffic lights" evaluation system [53], adding an additional qualitative system that consists of giving a letter to each color, with the letter C corresponding to the color red, B to yellow, and A to green. The range of scores for each letter and color varies according to the area.

On the other hand, to obtain the total score for each municipality, each area will have the same importance, for the same reason that occurs with the indicators; in other words, as the index is based in the law, this is giving the same importance to all the areas of evaluation. The score can be found as follows:

$$BTImunicipality = \frac{\sum_{=a}^{f} BTIarea_i}{6}$$

where *municipality$_j$* is the name of the municipality evaluated and *area$_i$* represents the area of information which is evaluated (from *a* to *f*, shown above).

In this way, the score for each municipality goes from 0 to 100, establishing a range of scores with identifying letters and colors in order to show more visual and unequivocal results. To this end, taking as a starting point the color assignment made in the Infoparticipa Map, the "traffic lights" evaluation system [54] has been followed and has been established as follows (Table 1):

**Table 1.** Ranges of scores and corresponding letters and colors.

| Range | Letter | Color |
|---|---|---|
| 0–49.99 | C | Red |
| 50–74.99 | B | Yellow |
| 75–100 | A | Green |

Source: The authors.

Thus, a graduation in compliance with the transparency law is established, from those municipalities that do not comply with the law (letter C) to those municipalities that fully comply with it (letter A), passing through municipalities that comply with the law but present deficiencies that mean that the law is not fully complied with (letter B).

*3.2. Methodological Application: Comparison between BTI, ITA, DYNTRA, and Infoparticipa Map: Data and Sample*

The main objective of this paper is to check the differences in transparency of municipalities between the ITA, DYNTRA, the Infoparticipa Map, MESTA indices, and the BTI. For this, an evaluation is to be carried out with the BTI on a sample of municipalities that is composed of the Spanish provincial capitals, and it will be compared with the rest of the indices with data from the year 2017, since that is the year in which the municipalities had to comply with the transparency objective because it was the end of the extension granted by the law. In addition, it is the year for which there are the most data available for the indices to be compared.

Based on this, we are governed by Article 121 of Law 7/1985, Regulating the Bases of Local Regime (incorporated by Law 53/2003, on measures for the modernization of local government) in municipalities classified as "large population", which are those that meet some of the following characteristics: (a) population over 250,000 inhabitants; (b) provincial capitals with population over 175,000 inhabitants; and (c) provincial capitals, autonomic capitals, or headquarters of autonomous institutions.

Thus, the sample is composed of the 52 Spanish cities that are provincial capitals, including the autonomous cities of Ceuta and Melilla; the main reason for choosing this sample is that they are cities that the rest of the indices studied have evaluated, so they can be compared at the same level.

As for the evaluation of each provincial capital for the BTI, the information required for each indicator is obtained from the transparency portals of each of the capitals, going if necessary to the generic website in case the information is not found on that portal.

## 4. Results

With the data obtained from the BTI indicator based on the evaluation of the 52 provincial capitals, it is highlighted that in 2017, the year of consultation of the indicators for the BTI, there were provinces that did not have a transparency portal, but currently all of them have already established these portals.

This leads us to consider that, when consulting the indicators of the different transparency indices, it is now much easier to search for the information, whereas years ago it was necessary to obtain it outside the portals since they were not available.

With respect to the research of the information of the MESTA index in 2017, an evaluation was only developed for 10 city councils, two deputations, and one commonwealth, but the individual assessment of each case was not made public [55]. In this sense, the MESTA index has been impossible to compare with the rest of the indices, and the study of the present work has been carried out with the other indices, since information for them was available.

It is also necessary to mention that, regarding the evaluation of the DYNTRA index, there are provincial capitals that have not been evaluated since 2015 or for which no data are obtained for the year under investigation, which limits the efficient analysis of the research.

Thus, in the evaluation performed, the results obtained are shown in Table 2.

**Table 2.** Results obtained in the evaluation with BTI.

| MUNICIPALITY | Letter | Color | BTI | MUNICIPALITY | Letter | Color | BTI |
|---|---|---|---|---|---|---|---|
| Madrid | A | Green | 86.39 | Soria | B | Yellow | 60.56 |
| Huelva | A | Green | 83.61 | Toledo | B | Yellow | 59.72 |
| Oviedo | A | Green | 80.28 | Zaragoza | B | Yellow | 59.72 |
| Salamanca | A | Green | 76.39 | Huesca | B | Yellow | 59.44 |
| Cuenca | B | Yellow | 74.44 | Pontevedra | B | Yellow | 59.44 |
| Córdoba | B | Yellow | 73.89 | Málaga | B | Yellow | 57.78 |
| Vitoria-Gasteiz | B | Yellow | 73.33 | Guadalajara | B | Yellow | 56.94 |
| Barcelona | B | Yellow | 72.78 | Palencia | B | Yellow | 56.11 |
| León | B | Yellow | 72.22 | Albacete | B | Yellow | 55.83 |
| Ourense | B | Yellow | 72.22 | Burgos | B | Yellow | 54.72 |
| Logroño | B | Yellow | 71.11 | Jaén | B | Yellow | 54.72 |
| Donostia | B | Yellow | 70.83 | Melilla | B | Yellow | 54.44 |
| Santander | B | Yellow | 70.83 | Lugo | B | Yellow | 53.33 |
| Sevilla | B | Yellow | 70.83 | Teruel | B | Yellow | 53.06 |
| Valladolid | B | Yellow | 69.44 | Santa Cruz de Tenerife | B | Yellow | 52.50 |
| Palma de Mallorca | B | Yellow | 68.89 | Bilbao | B | Yellow | 51.67 |
| Granada | B | Yellow | 67.22 | Girona | B | Yellow | 50.56 |
| Castellón de la Plana | B | Yellow | 66.67 | Cáceres | B | Yellow | 50.56 |
| Ciudad Real | B | Yellow | 66.39 | Zamora | B | Yellow | 50.00 |
| Valencia | B | Yellow | 66.39 | Segovia | C | Red | 48.06 |
| Murcia | B | Yellow | 63.89 | Badajoz | C | Red | 46.11 |
| Pamplona/Iruña | B | Yellow | 63.33 | A Coruña | C | Red | 45.28 |
| Lleida | B | Yellow | 62.78 | Ceuta | C | Red | 44.44 |
| Alicante | B | Yellow | 62.50 | Cádiz | C | Red | 41.11 |
| Las Palmas de Gran Canaria | B | Yellow | 61.11 | Ávila | C | Red | 40.56 |
| Tarragona | B | Yellow | 60.83 | Almería | C | Red | 25.00 |

Source: The authors.

As can be observed, none of the municipalities evaluated manages to reach a score of 100%, as happens in the DYNTRA index; however, in the ITA, 13 municipalities reach 100%, as does one municipality in the Infoparticipa Map index. It is worth highlighting the city of Almería, as it scored the worst in terms of information transparency due to the

fact that, in 2017, it did not have a transparency portal and the information through other means was not very accessible and even unavailable.

Regarding the degree of compliance with the Transparency Law, we observe that only 7.41% of the provinces fully comply with the law; 77.77% comply with it, but with deficiencies. However, 14.81% do not comply with it, which leads us to conclude that there is still much information that is not accessible to citizens and that we must continue to have an impact on this aspect in order to achieve transparency in all municipalities.

Having commented on the main results of the BTI, Table 3 shows the most noteworthy results in a descriptive analysis of all the indices.

**Table 3.** Average, maximum, and minimum, scores of the provincial capitals in each index.

| Index | Average% | Max% | Min% |
|---|---|---|---|
| BTI | 60.97 | 86.39 | 25 |
| ITA | 89.56 | 100 | 40.6 |
| DYNTRA | 64.46 | 90 | 33.13 |
| Infoparticipa Map | 64.27 | 100 | 30.77 |

Source: The authors.

As can be observed, it is possible to see how the evaluation of the BTI, on average, is more demanding than the rest of the indices, since the average score of the BTI is 60.97, compared to the 89.56 points of the ITA 2017 index, 64.46 of DYNTRA, and 64.27 of the Infoparticipa Map. In this sense, the average scores of the DYNTRA and the Infoparticipa Map indices may denote some similarity to the BTI in terms of measuring information transparency, but all agree that most municipalities should improve their transparency portal.

Regarding the maximum score obtained by each of the indices, it is observed that only the ITA and the Infoparticipa Map manage to reach a score of 100% with respect to the fulfilment of their indicators, compared to the DYNTRA index, whose maximum is 90%, and 86.39% for the BTI. It is necessary to point out with respect to the Infoparticipa Map that only one municipality in the sample has been awarded the maximum score, while the ITA awards it to 13 municipalities.

As for the minimum score obtained by each provincial capital, 25% is the lowest score in the BTI, with 30.77%, 33.13%, and 40.6%, being the lowest scores for the Infoparticipa Map, DYNTRA, and ITA indicators, in that order.

In this way, it is possible to appreciate that the ITA 2017 index stands out from the other indices, both for having a higher minimum than the rest and for having a greater number of municipalities with the maximum score, with which it could be confirmed that the contents to be evaluated within each indicator are known in advance by the municipalities evaluated, causing public managers to focus on such contents and adapt the information published to this index. This fact makes one think that the transparency portals have been developed to comply with said index exclusively, and not with the law.

After evaluating the BTI and the rest of the indices, a comparison will be made between the results obtained by our index and those published in the ITA, DYNTRA, and Infoparticipa Map, taking into account the limitation that some indices do not have scores for some of the provincial capitals. The comparative scores between the four transparency indices are shown in the following tables (Tables 4 and 5):

**Table 4.** Comparison of scores with the ITA, DYNTRA, and Infoparticipa Map (1).

| MUNICIPALITY | BTI | ITA | DYNTRA | INFOPARTICIPA |
|---|---|---|---|---|
| Madrid | 86.39 | 100 | 84.91 | 100 |
| Huelva | 83.61 | 100 | 51.7 | 75 |
| Oviedo | 80.28 | 100 | 52.67 | 75 |
| Salamanca | 76.39 | 93.8 | 54.09 | 67.31 |
| Cuenca | 74.44 | 86.9 | 36.18 | 53.85 |
| Córdoba | 73.89 | 86.3 | 60.38 | 55.75 |
| Vitoria | 73.33 | 100 | 90 | 69.23 |
| Barcelona | 72.78 | 100 | 84.28 | 98.08 |
| León | 72.22 | 100 | 77.5 | 51.92 |
| Ourense | 72.22 | 71.9 | 65.33 | 42.31 |
| Logroño | 71.11 | 100 | 83.33 | 73.08 |
| Donostia | 70.83 | 95.6 | 84.31 | 75 |
| Santander | 70.83 | 96.9 | 45.33 | 65.38 |
| Sevilla | 70.83 | 93.8 | 74.21 | 59.62 |
| Valladolid | 69.44 | 98.8 | 72.5 | 53.85 |
| Palma de Mallorca | 68.89 | 98.8 | - | 86.54 |
| Granada | 67.22 | 82.5 | 73.2 | 67.31 |
| Castellón de la Plana | 66.67 | 91.3 | - | 69.23 |
| Ciudad Real | 66.39 | 97.5 | - | 69.23 |
| Valencia | 66.39 | 90 | 70.51 | 69.234 |
| Murcia | 63.89 | 89.4 | - | 75 |
| Pamplona | 63.33 | 83.8 | 78 | 61.54 |
| Lérida | 62.78 | 100 | - | 98.08 |
| Alicante | 62.5 | 100 | 64.15 | 73.08 |
| Las Palmas de Gran Canaria | 61.11 | 100 | 70 | 59.62 |
| Tarragona | 60.83 | 72.5 | - | 84.62 |

Source: The authors.

This first table shows the best scores obtained by the provincial capitals in the BTI, with Madrid in first place with 86.39%, followed by Huelva, Oviedo, Salamanca, and Cuenca, with scores of 83.61%, 80.28%, 76.39%, and 74.44%, respectively.

In the ITA, there are 13 municipalities that obtain a perfect score in terms of information transparency in city councils. These capitals are Madrid, Huelva, Oviedo, Vitoria, Barcelona, León, Logroño, Lérida, Alicante, Las Palmas de Gran Canaria, Soria, Bilbao, and Cáceres. The first three provincial capitals coincide with the BTI, but there is a great difference in the results with respect to the evaluation of the rest of these municipalities, and it can be observed that municipalities with the highest score in ITA achieve little more than 60% in the BTI or other indices.

In the case of the DYNTRA index, Vitoria, Bilbao, Zaragoza, Madrid, and San Sebastian, have the highest scores, with percentages of 90%, 87.42%, 85.63%, 84.91%, and 84.31%, respectively.

In the evaluation of the Infoparticipa Map index, the five municipalities with the highest percentage of transparency are Madrid with 100%; Barcelona, Gerona, and Lérida, with 98.08%; and Toledo with 90.38%.

From these data, it can be concluded that Madrid, for all the indices, is the provincial capital that offers the most information on transparency to citizens in its portal, coinciding with the ITA and Infoparticipa Map, in that Madrid complies with the law at 100% or, as in the case of DYNTRA and BTI, complies with it, but with deficiencies.

On the other hand, it can be observed that there are certain similarities between DYNTRA, the Infoparticipa Map, and BTI, and there is a clear difference with respect to ITA.

**Table 5.** Comparison of scores with the ITA, DYNTRA, and Infoparticipa Map (2).

| MUNICIPALITY | BTI | ITA | DYNTRA | INFOPARTICIPA |
|---|---|---|---|---|
| Soria | 60.56 | 100 | - | 46.15 |
| Toledo | 59.72 | 58.1 | 57.5 | 90.38 |
| Zaragoza | 59.72 | 94.4 | 85.63 | 75 |
| Huesca | 59.44 | 95.6 | - | 80.77 |
| Pontevedra | 59.44 | 78.1 | 65.33 | 63.46 |
| Málaga | 57.78 | 89.4 | 80.79 | 65.38 |
| Guadalajara | 56.94 | 99.4 | - | 69.23 |
| Palencia | 56.11 | 89.4 | - | 53.85 |
| Albacete | 55.83 | 95 | - | 55.77 |
| Burgos | 54.72 | 88.1 | 72 | 59.62 |
| Jaén | 54.72 | 58.1 | 39.62 | 50 |
| Melilla | 54.44 | - | - | 40.38 |
| Lugo | 53.33 | 71.9 | 54.67 | 55.77 |
| Teruel | 53.06 | 83.8 | 56.6 | 61.54 |
| Santa Cruz de Tenerife | 52.5 | 87.5 | 49.33 | 51.92 |
| Bilbao | 51.67 | 100 | 87.42 | 65.38 |
| Cáceres | 50.56 | 98.8 | 54.67 | 53.85 |
| Gerona | 50.56 | 100 | 57.05 | 98.08 |



| | | | |
|---|---|---|---|
| Zamora | 50 | 97.5 | 52.98 | 65.38 |
| Segovia | 48.06 | 90.6 | - | 42.31 |
| Badajoz | 46.11 | 75.6 | 44.52 | 65.38 |
| A Coruña | 45.28 | 83.1 | 66.67 | 51.92 |
| Ceuta | 44.44 | - | - | 34.62 |
| Cádiz | 41.11 | 86.9 | 54.72 | 42.31 |
| Ávila | 40.56 | 86.3 | - | 44.23 |
| Almería | 25 | 40.6 | 33.13 | 30.77 |

Source: The authors.

This second table shows the 26 provincial capitals with the lowest scores for the BTI, of which Almería, with 25%, is the municipality with the lowest level of transparency for each of the indicators evaluated. Ávila with 40.56%, Cádiz with 41.11%, Ceuta with 44.44%, and A Coruña with 45.28%, are the ones that follow Almería with the lowest scores. A fact that is striking in these last four cities is that in the ITA, their scores range between 83% and 86%, and Almería, Jaén and Toledo, with 40.60% and 58.1%, respectively, are the cities with the lowest scores.

As for the DYNTRA index, Almería, Cuenca, Jaén, Badajoz, and Santander, are five of the municipalities that do not comply with the Transparency Law, obtaining a score below 50%. Finally, Almería with 30.77%, Ceuta with 34.62%, Melilla with 40.38%, and Ourense, Segovia, and Cádiz with 42.31%, are, for the Infoparticipa Map, those with the lowest scores.

After this evaluation, it is concluded that Almeria is the city that offers the least information on transparency for all indices, confirming the need for accessibility to information on the portal. In addition, the difference between ITA and the rest is clearly seen again, placing in this part of the list municipalities that reach between 90% and 100% according to their criteria. It is also worth noting that the Infoparticipa Map and BTI are the only indices that have evaluated the Autonomous Cities of Ceuta and Melilla, neither of them obtaining a percentage higher than 55%, which shows that these portals need to be updated and more emphasis needs to be placed on them in order to be evaluated by the rest of the indices.

To obtain a clearer view of the above results, two figures are shown below, one for each of the 26 provincial capitals, showing the difference between scores. In general, it is observed that ITA has an overvaluation in most of the municipalities (Figures 1 and 2).

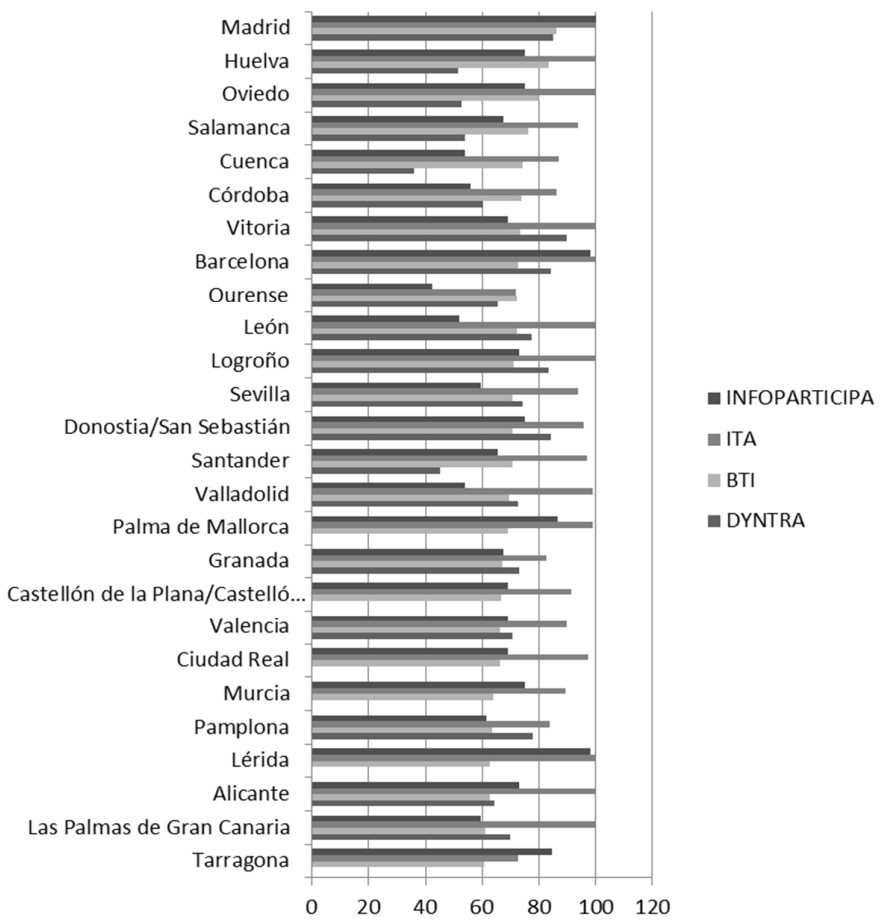

**Figure 1.** Scores of BTI, ITA, DYNTRA, and INFOPARTICIPA Map, and differences (1). Source: The authors.

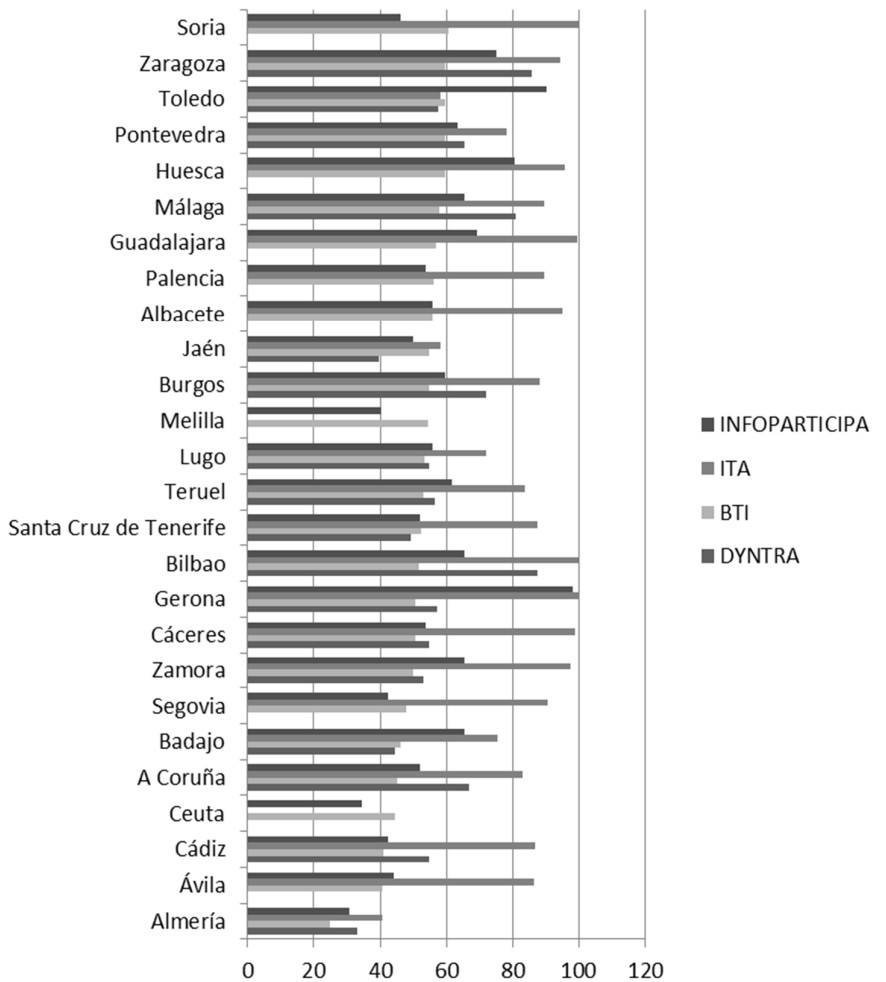

**Figure 2.** Scores of BTI, ITA, DYNTRA, and INFOPARTICIPA Map, and differences (2). Source: The authors.

## 5. Conclusions

The different socio-political crises that have occurred in recent years have led to a growing demand from the population for public information. The need to promote transparency in public administrations and, therefore, to increase public confidence has given rise to the development of both the concept itself and the regulations governing it. In recent years, these administrations have been obliged to seek various alternatives to evaluate their degree of transparency. Among these entities are the municipalities, whose commitment to transparency must be greater, since they are the bodies whose actions most directly affect citizens.

In this context, this paper has presented a new transparency index called BTI, developed by [49] and whose objective is to measure in the best possible way the degree of compliance of Spanish city councils in line with Law 19/2013, giving greater depth and breadth when measuring each of the indicators. In relation to the breadth, the BTI is composed of 20 indicators distributed in six different valuation areas, which have the same relative importance, and each of which has the same weight in the valuation due to the fact that the transparency law itself gives the same importance to each obligation to be fulfilled. The main novelty of this index is that it has two dimensions: breadth, which takes into account the number of indicators and the minimum information to disclose; and depth, which takes into account the quality of the information. Each of the 20 indicators

has a score ranging from 0 to 3, which increases according to the degree of compliance with the established requirements. In addition, a "traffic lights" system has been implemented to assign a specific color to each score obtained. Finally, in both the results by area and the total result, depending on the score obtained, a certain letter and color are assigned, which are established by score ranges.

Likewise, this work aims to compare the BTI with three other transparency indices: ITA, DYNTRA, and the Infoparticipa Map. For this purpose, a sample of 52 provincial capitals has been chosen, including the autonomous cities of Ceuta and Melilla, in order to evaluate the results obtained in the different indices and to compare them with the index developed in this work.

Regarding the application of the BTI, none of the municipalities analyzed obtained the maximum score of 100%, with Huelva and Madrid achieving the highest scores. The same does not occur with the ITA, where 13 municipalities obtain the highest possible score. As for the DYNTRA index and the Infoparticipa Map, the municipalities that achieve a higher value vary with respect to the previous indices. In the case of DYNTRA, the municipalities of Vitoria and Bilbao have the highest score, although none of them reach 100%, while in the Infoparticipa Map it is Barcelona, Gerona, Lérida, and once again Madrid, being the only municipality that reaches 100%.

In average terms, it can be concluded that the BTI is more demanding than the rest, since the average score barely exceeds 60 points, while DYNTRA and the Infoparticipa Map obtain higher scores, although similar. However, ITA stands out with an average score of almost 90 points, which reflects that the transparency portals are designed very much in line with this index, and that the indicators, in addition to being dichotomous, are known beforehand, so they focus on them to achieve a good score. This is especially noteworthy when trying to obtain information related to aspects other than those evaluated by the ITA, since in many cases there is little or no information. These results lead us to the conclusion that, in three of the four indices analyzed, most of the municipalities are limited to complying with the transparency law, although only some of them comply with a high rating.

Public administrations are increasingly aware of the importance of being transparent with their citizens, so the dedication and effort made by these administrations is reflected through the publication of information on municipal websites. Although more work is needed for these institutions to achieve real transparency, the information published by public administrations on their web portals has become a strategic and useful tool for citizens and a way to improve the legitimation of the decisions taken and the efficiency of the services provided [23,54].

In this context, this research work has also made two contributions: on the one hand, it can be a reference for Spanish readers to compare indices and observe the differences of measuring transparency with one concrete index or another; and on the other hand, it can be a reference point to obtain the same comparison in other countries where there is an index of reference (as ITA in Spain), to measure the differences with other existing indices adapted to the regulation of that countries. In this sense, the results of future works should observe significant differences between indices (see [56,57]).

However, a series of limitations were encountered when collecting information for the preparation of this paper. The latest information available and on which it was possible to work for most of the indices analyzed dates from 2017, so it was not possible to access more recent data. In addition, when consulting data prior to that date, it was more difficult to obtain the required information for certain municipalities, as they do not have transparency portals on their websites. Another limitation arose with the DYNTRA index, where provincial capitals that have not been evaluated since 2015 can be found, so information relating to later years cannot be obtained. On the other hand, we intended to analyze the Transparency Evaluation and Monitoring Methodology, or MESTA; however, no information has been found in this regard for the study period, so such analysis had to be discarded from the present work. Finally, another limitation faced when we applied the

new index was the difficulty to find the information that fulfil some indicators, as the majority of the portals were designed following the same structure of the indicators of ITA, making the process much less accessible and only good to obtain a high score on that index. These limitations can be solved if all the indices present their results and make them openly available for society [58,59]

To improve the aforementioned, it is necessary to make transparency indices dynamic instruments that allow for the analysis of relevant information. Therefore, it is necessary to continue investigating in this field of study. We propose as possible future lines of research the analysis of different municipalities according to a series of variables [60], such as the number of inhabitants or their geographic location, in order to compare, according to these variables, how the information provided in their corresponding transparency portals evolves.

**Supplementary Materials:** The following supporting information can be downloaded at: https://www.mdpi.com/article/10.3390/su142013081/s1, File S1: ANNEX 1: Sample of indicators of the BTI index.

**Author Contributions:** Conceptualization, E.Z.-G.; Methodology, M.G.-M.; Resources, G.L.-P.; Supervision, J.-C.G.-R.; Writing – original draft, M.G.-M., G.L.-P. and E.Z.-G.; Writing – review & editing, J.-C.G.-R. All authors have read and agreed to the published version of the manuscript.

**Funding:** This research was funded by Ministry of Science, grant number PID2021-128713OB-I00 and grant number P20_00605.

**Institutional Review Board Statement:** Not applicable.

**Informed Consent Statement:** Not applicable.

**Data Availability Statement:** The data presented in this study are available on request from the corresponding author.

**Conflicts of Interest:** The authors declare no conflicts of interest. The funders had no role in the design of the study; in the collection, analyses, or interpretation of data; in the writing of the manuscript; or in the decision to publish the results.

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
