# Peer review of "The Importance of Measuring Local Governments’ Information Disclosure: Comparing Transparency Indices in Spain"

_sustainability, doi:10.3390/su142013081_

Round 1

Reviewer 1 Report

The article offers a new transparency index for spanish municipalities. 

A few concerns:

1) try to better justify in what your index is better than the existing ones

2) it would be nice to be able to read all your indicators perhaps in an addendum to compare them to the indicators of the existing index. have you fond some dimensions or variables previously neglected? it is not clear what such indicators deal with? only the website? 

3) try to better justify the choice of non using weights among the indexes and among the areas of evaluations. In my view, weights are necessary since variables are not similar in importance;

4) there is no theoretical contribution whatsoever, nor an attempt to engage with the existing literature. i.e. do you believe that stakeholders should not have a voice in evaluating transparency? I believe that in time of co-creation, co-evaluation and so on, they should

5) finally, there is a vast international literature on transparency. don't rely only on Spanish authors. 

Author Response

We acknowledge the comments proposed to review our manuscript. We made the necessary changes in order to improve it. Please see the attachment for more detailed information

Reviewer 2 Report

I will recommend to modify the paper as follows: 1) Abstract should be written as 1) objective, methodology, 3) findings, 4) conclusion, and 5) implications.

Introduction: The authors present the motivation and problems in section introducción, but lack a paragraph that indicates "our scientific contribution is ....."

Theoretical Approach: the theoretical approach is well described. The authos describe a review good of the transparency concept as well as the development of the transparency act in Spain. The point 2.3  How can transparency be measured is clear.

I suggest in this section to put current appointments (last 7 years), you see quotes from 20 years ago.

Data and Methodology: Are cleary describe methodology to measure the degree of transparency of a municipality: the Bidimensional Transparency Index and also Methodological application.

Results: The authors are  indices analysed show that most municipalities only fulfil the Transparency Act at the minimum level, with only a few municipalities reaching the maximum rating range.

Author Response

(The authors gave the same response as above.)

Reviewer 3 Report

The article addresses a current and sensitive topic and it is well structured. Transparency regarding the activity of public institutions is a coordinate for a democratic society. The results of this research are presented appropriately. In the manuscript, the authors propose a new index for evaluating the transparency of a municipality.

However, there are some areas that deserve improvement:

1.   The section 2, Theoretical Approach, could be revised with references to several recently published papers. Such a current topic requires new bibliographic resources.

2.   The following passage is found in a slightly different form in the cited article:

the online publicity of all the acts of government and their representatives to provide civil society with relevant in formation in a complete, timely, and accessible manner” (Lines 74-76)

The original quote is:

”the publicity of all the acts of government and its representatives to provide civil society with relevant information in a timely, useful and comparable way and in an accessible format” (in Da Cruz, N. F., Tavares, A. F., Marques, R. C., Jorge, S., & De Sousa, L. (2016). Measuring local government transparency. Public Management Review, 18(6), 866-893.)

3.   I consider that the English wording for this passage could be taken:

“Everyone has the right … to investigate and receive information” (lines 137-138)

The original quote is:

” Everyone has the right … to seek, receive and impart information”.

(https://www.un.org/en/about-us/universal-declaration-of-human-rights)

4.   The formulas (L323, L334) must be accompanied by the definition of the symbols used. In the formula for determining BTI municipality, the index of summation must be noted differently (not i).

5.   In the Conclusions section, I consider that some limitations of the methodology for calculating the transparency index proposed by the authors should be highlighted, other than those that occurred when the information was collected.

6.   The first article in the bibliography should be cited in the format:

Magdaleno, M. L. A., & García-García, J. (2014). Evaluación de la transparencia municipal en el Principado de Asturias. Auditoría pública, 64, 75-86.

Author Response

(The authors gave the same response as above.)

Round 2

Reviewer 1 Report

The paper has improved. 

Author Response

We want to appreciate all the comments made by the reviewer, making our work more robust to publish in the journal.